# Assessing Tumour Haemodynamic Heterogeneity and Response to Choline Kinase Inhibition Using Clustered Dynamic Contrast Enhanced MRI Parameters in Rodent Models of Glioblastoma

**DOI:** 10.3390/cancers14051223

**Published:** 2022-02-26

**Authors:** Sourav Bhaduri, Clémentine Lesbats, Jack Sharkey, Claire Louise Kelly, Soham Mukherjee, Arthur Taylor, Edward J. Delikatny, Sungheon G. Kim, Harish Poptani

**Affiliations:** 1Centre for Preclinical Imaging, Department of Molecular and Clinical Cancer Medicine, University of Liverpool, Liverpool L69 3BX, UK; sourav.bhaduri@liverpool.ac.uk (S.B.); clementine.lesbats@icr.ac.uk (C.L.); jack.sharkey@perkinelmer.com (J.S.); claire.kelly@liverpool.ac.uk (C.L.K.); soham.mukherjee@liverpool.ac.uk (S.M.); 2Division of Radiotherapy and Imaging, The Institute of Cancer Research, London SM2 5NG, UK; 3Department of Molecular Physiology & Cell Signalling, University of Liverpool, Liverpool L69 3BX, UK; taylora@liverpool.ac.uk; 4Department of Radiology, University of Pennsylvania, Philadelphia, PA 19104, USA; delikatn@pennmedicine.upenn.edu; 5Department of Radiology, Weill Cornell Medical College, New York, NY 10021, USA; sgk4001@med.cornell.edu

**Keywords:** animal model, dynamic contrast-enhanced MRI, pharmacokinetic models, intra-tumoral heterogeneity, glioblastoma, clustering, choline kinase, JAS239

## Abstract

**Simple Summary:**

This study was designed to monitor changes in DCE-MRI-based parameters in preclinical GBM models in response to choline kinase inhibition using a cluster analysis approach. In terms of therapeutic response in F98 rat GBMs, a sustained decrease in permeability and perfusion and increased necrosis was observed during treatment with JAS239 as compared to control animals. No significant differences in these parameters were found for the GL261 mice GBMs. The study demonstrates that region-based clustered pharmacokinetic parameters obtained using DCE-MRI can be used for detecting and assessing tumour haemodynamic heterogeneity, which may be useful in assessing therapeutic response.

**Abstract:**

To investigate the utility of DCE-MRI derived pharmacokinetic parameters in evaluating tumour haemodynamic heterogeneity and treatment response in rodent models of glioblastoma, imaging was performed on intracranial F98 and GL261 glioblastoma bearing rodents. Clustering of the DCE-MRI-based parametric maps (using Tofts, extended Tofts, shutter speed, two-compartment, and the second generation shutter speed models) was performed using a hierarchical clustering algorithm, resulting in areas with poor fit (reflecting necrosis), low, medium, and high valued pixels representing parameters $K^{trans}$, $v_{e}$, K_ep,_ $v_{p}$_,_ $\tau_{i}$ and $F_{p}$. There was a significant increase in the number of necrotic pixels with increasing tumour volume and a significant correlation between $v_{e}$ and tumour volume suggesting increased extracellular volume in larger tumours. In terms of therapeutic response in F98 rat GBMs, a sustained decrease in permeability and perfusion and a reduced cell density was observed during treatment with JAS239 based on $K^{trans}$, $F_{p}$ and$\;v_{e}$ as compared to control animals. No significant differences in these parameters were found for the GL261 tumour, indicating that this model may be less sensitive to JAS239 treatment regarding changes in vascular parameters. This study demonstrates that region-based clustered pharmacokinetic parameters derived from DCE-MRI may be useful in assessing tumour haemodynamic heterogeneity with the potential for assessing therapeutic response.

## 1. Introduction

Glioblastoma multiforme (GBM) is the most prevalent and fast-growing primary brain tumour with poor prognosis and a median survival rate of 14–16 months after diagnosis [1]. It is more heterogeneous and hypoxic compared to other types of brain tumours [2]. As survival with GBM is short, it is critical to determine the efficacy of therapy early on during treatment. Early interventions can be made in case a therapeutic strategy is ineffective. Technological developments in magnetic resonance imaging (MRI) have helped to provide new insights into the diagnosis, classification, and understanding of tumour biology [3,4]. Perfusion imaging provides multiple parameters related to the blood volume and flow, as well as vascular endothelial leakiness. The most common perfusion imaging techniques involving the administration of contrast agent are dynamic susceptibility contrast imaging (DSC), and dynamic contrast-enhanced imaging (DCE). Perfusion-weighted MRI techniques are used increasingly in assessing GBMs; most tumour imaging protocols now include either DCE and/or DSC imaging [5]. DSC is a prevalent method in clinical neuro-oncology, but is sensitive to susceptibilities and gives inaccurate estimates when dealing with leaky and tortuous capillaries in tumours [6]. Leakage correction plays an important role to compensate for this [7,8]. DCE-MRI can characterise vascular permeability in tumours and has an advantage over DSC-MRI due to its greater signal-to-noise ratio and spatial resolution, although imaging acquisition time is relatively longer [6].

DCE-MRI has been used for clinical brain tumour imaging [9]. It can be processed using several pharmacokinetic models to quantify the kinetics of contrast agents crossing the blood–brain barrier. These models have been used to quantify physiologically relevant parameters, including estimation of volume transfer constant ($K^{trans}$, measured in min^−1^) of gadolinium-based contrast agent from the intravascular compartment to tumour interstitium [10]. $K^{trans}$ has also been used for predicting short-term response [11,12] and overall survival in head and neck cancer patients [13]. The diagnostic ability of other DCE-MRI parameters such as volume of extravascular extracellular space ($v_{e}$), rate constant between tumour interstitium and blood plasma (K_ep,_ measured in min^−1^), plasma volume ($v_{p}$) have also been reported in glioma patients [14,15]. In addition, the mean intracellular water lifetime ($\tau_{i}$, measured in sec), has been suggested to be a reliable marker of cellular energy turnover [16]. Plasma flow ($F_{p}$, measured in mL/100 g/min) correlates with tumour oxygenation and, thus, provides more specific information on radiotherapy efficacy [17].

GBMs are formed by tumour cells, which differ in their morphology, genetics and biological behaviour [18,19]. They are typically heterogeneous both on genetic and histo-pathological levels, with intratumoural spatial variation in the cellularity, angiogenesis, extravascular extracellular matrix, and areas of necrosis [20,21]. The heterogeneity can be with regards to cellular morphology, gene expression, metabolism, and angiogenic and proliferative potential, some of which can be investigated using DCE-MRI, for example, in investigating angiogenic processes [22,23]. Substantial intra-tumour heterogeneity correlates with increased morbidity, mortality, and recurrence rates in patients [24]. This heterogeneity results in non-uniform distribution of tumour-cell subpopulations within disease sites [25]. Thus, an accurate assessment of tumour haemodynamic heterogeneity is essential for developing effective therapies. The validity of different DCE-MRI models to assess tumour haemodynamic heterogeneity has not been studied in detail. A recent study identified physiological tumour habitats from DCE-MRI data using parameters derived from Tofts model and evaluated their alterations in response to therapy in preclinical breast cancer models [26]. Although $K^{trans}$, $v_{e}$, $v_{p}$, K_ep_ can be estimated using the commonly used general kinetic or Tofts (GKM) or its extended version (ETM). Other models, such as the shutter speed (SSM) and two-compartment exchange model (2CXM), can provide additional physiologically sensitive parameters such as $\tau_{i}$ and $F_{p}$, respectively. This study aims to assess tumour haemodynamic heterogeneity using the parameters derived from these models.

There is an urgent need to evaluate new drugs for the treatment of GBMs. Recent studies have explored the possibility of altering the expression or activity of enzymes involved in choline metabolism as a novel therapeutic target for cancer treatment. A recent study demonstrated that ^1^H MR spectroscopy (MRS) can be used to detect a decrease in total choline (tCho) that is associated with the inhibition of choline kinase (ChoK) activity by MN58b in gliomas [27]. The inhibition of ChoK may help sensitise GBMs to chemotherapy and radiotherapy [27]. Another ChoK inhibitor, JAS239, also inhibits ChoK intracellularly, preventing choline phosphorylation, and has been shown to induce tumour growth arrest and cell death in a breast cancer model [28,29]. Assessment of tumour haemodynamic heterogeneity could identify the sub population of cells that are not perfused adequately and, therefore, are more likely to be resistant to treatment. It is important for therapeutics to be directed to specific tumour subvolumes. Thus, another goal of this study was to monitor changes in tumour haemodynamic heterogeneity by quantifying the changes in the pharmacokinetic parameters, in response to treatment with the ChoK inhibitor, JAS239.

## 2. Materials and Methods

To assess the generalizability of our methods, we performed the imaging studies on two syngeneic rodent models of GBM, the F98 GBM in rats and a GL261 models in mice, as both have been shown to recapitulate several features of human GBM and have been used previously in the DCE-MRI literature [2,3,30,31].

F344 Fischer rats (n = 34) were implanted with 50,000 F98 GBM cells and C57BL6 mice (n = 10) were implanted with 500,000 GL261 GBM cells. Intracranial tumours were developed by transcranial injection of the GBM cells in the right cortex. The rodents were secured on a stereotaxic frame, a burr hole was drilled through the skull 3 mm right for rats (1.5 mm right for mice) and 3 mm posterior for rats (2 mm posterior for mice) from the bregma, and GBM cells suspended in 5 µL serum-free medium were injected 2.5 mm into the brain for rats (2 mm for mice).

The MR images were obtained using a 9.4 T Bruker Biospec scanner (Bruker BioSpin, Ettlingen, Germany). FAIR (flow sensitive alternating inversion recovery) pulse sequence using non-selective 180° pulse and 40 inversion times varying from 100 to 7900 ms in steps of 200 ms with matrix size = 128 × 128, FOV of 30 × 30 mm^2^ and slice thickness = 1.16 mm, was used to acquire the pre-contrast T1 maps from a set of six F98 tumour bearing rats. A 3D FLASH sequence was used to obtain T_1_-weighted images with five flip angles of 2°, 5°, 7°, 12° and 15° and MRI scan parameters: FOV = 20 × 20 × 4.8 mm^3^; matrix size = 128 × 64 × 8; TR/TE of 15/1.5 ms, from all GL261 GBM bearing mouse brains to obtain pre-contrast T1 maps.

Dynamic 3D multi-gradient-echo (MGE) sequence was used to record the kinetics of the contrast agent. For the dynamic imaging, a matrix size of 128 × 64 × 8, FOV of 30 × 30 × 9.28 mm^3^ (for rats) and 20 × 20 × 4.8 mm^3^ (for mice), TR/TE1/TE2 = 14/2.25/4.76 ms, and a 12° flip angle was used. A bolus of gadopentetate in saline at the standard dose of 0.1 mmol/kg for rats and 0.2 mmol/kg for mice (for better SNR) was injected through a tail vein catheter, starting at 1 min after collection of baseline images. 180 volumes were collected with a temporal resolution of 5.37 s per volume. Higher resolution anatomical T_2_- weighted images were also acquired using a RARE sequence (TR/Effective TE of 4167/33 ms, 0.3 mm slice thickness, FOV = 30 × 30 mm^2^, 256 × 256 matrix, Rare Factor = 8) for co-registration of the DCE images and tumour volume calculation.

The tumour haemodynamic heterogeneity was evaluated on the basis of change in tumour volumes using the different pharmacokinetic models (rats; n = 30, mice; n = 5).

For therapeutic response monitoring, a separate cohort (rats; n = 4, mice; n = 5) received JAS239 treatment (4 mg/kg/day injected intraperitoneally for 5 consecutive days), or saline (rats; n = 4, mice; n = 5). Animals were imaged on day 0 (*T0*, *baseline*), day 3 (*T3*, *during*), 6 (*T6*, *end*) of treatment and post-treatment day 8 (*T8*) for rats and day 10 (*T10*) for mice.

### 2.1. Data and Image Processing

$T_{2}^{*}$ correction was performed on the DCE datasets, utilising the data collected at two echo times (TE1 and TE2) to correct for signal loss due to field inhomogeneities arising from magnetic or tissue component susceptibility distortions [32]. T1 maps were generated by fitting the pixel-wise image intensities at different inversion times (rats; n = 6) and at different flip angles (mice; n = 10) using a non-linear least-square fitting ‘lsqcurvefit’ routine in MATLAB R2021a. DCE images were co-registered to the T_2_- weighted images using rigid body registration to correct for bulk motion. The region of interest (ROI) from brain slices containing tumour and the arterial input function (AIF) from the superior sagittal sinus were drawn manually on the co-registered DCE images. The mean T1 value in the tumour ROI from the six F98 rat GBMs was chosen as an initial T1 value for conversion of signal intensity to concentration curves. The ROI was also drawn in the tumour region across every slice manually on the T_2_- weighted images; the number of pixels in the ROI was multiplied with the pixel dimensions to get the tumour volume. The tumour volumes ranged from 0.006–0.04 cc with a mean value of 0.015 ± 0.008 cc.

$K^{trans}$, $v_{e}$ and K_ep_ were derived from Tofts model (TM) [33], extended Tofts model (ETM) [33], shutter speed model (SSM) [16,34], two-compartment exchange model (2CXM) [35,36] and the second generation shutter speed model (SSM2) [37], respectively. The $v_{p}$ was derived using ETM, 2CXM and SSM2 models, respectively [33,35,36,37]. $\tau_{i}$ was estimated using SSM and SSM2 models, respectively [16,34,37]. $F_{p}$ was estimated using the 2CXM model [35,36]. More details about these models can be found in the Appendix A. A non-linear least-square fitting ‘lsqcurvefit’ routine in MATLAB was used in all models. For model fitting, the initial parameter values were based on the literature values [38,39]. A hybrid bi-exponential and gamma variate model fitting of the AIF was performed prior to kinetic model analysis [40,41]. A population AIF was used whereby the AIF obtained from all the datasets were averaged after a selection-criteria based on area under each concentration curve (AUC), curve smoothness as described in [42,43]. The pharmacokinetic models used are typically not valid for necrotic tissue [44,45] and may result in poor fit in those regions. For all models, the goodness of fit (*R*^2^) values in the tumour ROI were within a range of 0.7–1.00. The values less than 0.7 resulted from noisy signals that did not show a pattern for contrast agent uptake in the tissues. Hence, all the parameters were set to zero in case of a poor fit (R^2^ < 0.7) resulting in a poor fit region or cluster reflecting necrosis.

A clustering analysis was used to assess regional changes in the tumour using a hierarchical clustering algorithm, a kind of agglomerative clustering. The algorithm was implemented in MATLAB. It uses a similarity measure to compare two vectors. Initially, each vector is considered as a single cluster. The clustering method uses a strategy to merge pairs of clusters. In each step of clustering, two clusters are merged until a threshold is achieved. A Ward’s linkage method and Euclidean distance measure was used [46]. Ward’s linkage method analyzes the variance of clusters and is the most suitable method for quantitative variables [46]. With hierarchical clustering, the sum of squares starts at zero (because every point is in its own cluster) and grows as the clusters are merged. Ward’s method keeps this growth as small as possible [46]. After creating the hierarchical tree of clusters, it is pruned by specifying an arbitrary number of clusters, which was set to 3 in this study (excluding the poor fit cluster). This was done to assess the tumour haemodynamic heterogeneity using a three-region clustering of the DCE-MRI-based parameters, resulting in areas of low, medium, and high values. The number of pixels in these cluster were indicated as $N_{x,y}$, where x = $K^{trans}$, $v_{e}$, K_ep,_ $v_{p}$_,_ $\tau_{i}$ and $F_{p}$ and y = low, med and high. The mean value of the parameters ($\bar{x})$ was also calculated across all the clusters.

### 2.2. Statistical Analysis

The goodness of fit was estimated using mean $R^{2}$ value (from all tumour pixels excluding the necrotic ones) for the different pharmacokinetic models. Pearson’s correlation coefficients were calculated to evaluate the correlation between tumour volumes and corresponding DCE-MRI derived pharmacokinetic parameters. The significance level was set at *p* ≤ 0.05. Differences between JAS239 and control group (% change in mean of estimated parameters on *T3*, *T6*, and *T8 or T10* scans with respect to *T0*) were evaluated using Wilcoxon rank sum test. All analyses were conducted using MATLAB. The pharmacokinetic models, showing a significant increase in the number of necrotic pixels with increase in tumour volume (based on Pearson’s correlation coefficient), along with a decrease in high $K^{trans}$ pixels with an increase in tumour volume (based on Pearson’s correlation coefficient) and also exhibiting high mean $R^{2}$ in the tumour ROI, were selected to assess response to treatment.

## 3. Results

The DCE-MRI data quality was good in all cases, and the tumour data were fitted well, as shown for the SSM model in a representative case (Appendix A). Figure 1 shows the four-region clustering of the DCE parameter maps (with SSM) performed using hierarchical clustering algorithm from tumour bearing rodents at three different days as the tumour grew (top to bottom).

Table 1 and Table 2 show the correlations between the clustered parameters and tumour volume from the 35 datasets. The 2CXM and SSM models provided the best fit (mean $R^{2}$ > 0.96) in the tumour ROI compared to the other models, as shown in Table 1B. These two models were, therefore, selected for further assessment of treatment response.

Representative example images from a JAS239-treated rat and a saline control are displayed in Figure 2A,B. The percentage change in volume and mean $K^{trans}$ (obtained using SSM) at time points *T0*, *T3*, *T6*, and *T8* are shown along with the treatment points marked with coloured arrows on the *X*-axis.

### 3.1. Necrotic Region

Table 1A shows the correlation values between tumour volume and necrotic regions (poorly fitted pixels) from the 35 datasets using different models. A significant correlation between tumour volume and the number of necrotic pixels was observed using the SSM (*r* = 0.53, *p* < 0.001) and the 2CXM (*r* = 0.54, *p* < 0.001) model.

For F98 rat GBMs, the SSM and 2CXM models demonstrated an increasing trend in the percentage of necrotic pixels due to JAS239 treatment from *T3* to *T8* compared to control animals (Figure 3A). However, this increase was only statistically significant on *T3* (*p* = 0.04).

There was an increasing trend in necrotic pixels with JAS239 treatment compared to control in GL261 cases (Figure 3B).
$$K^{trans}\;$$

As shown in Table 1 and Table 2, no significant correlation between tumour volume and clustered $K^{trans}$ pixels were observed using any of the models. Significant correlations were also not observed between mean $K^{trans}$ value and tumour volume using any models as shown in Appendix A.

For F98 rat GBMs, a significant decrease in $N_{K^{trans},high}$ was observed in control animals (*p =* 0.02) between *T3* and *T8*, respectively, using the 2CXM model, as shown in Figure 4. A significant decrease in $N_{K^{trans},high}$ (*p =* 0.02) with a significant increase in $N_{K^{trans},low}$ (*p =* 0.04) was observed on *T3* in JAS239-treated as compared to control animals using SSM. A significant decrease in $N_{K^{trans},high}$ (*p* = 0.02) was observed on *T6* in JAS239-treated compared to control animals using 2CXM. Appendix A shows the boxplots of percentage change in the mean $K^{trans}$ showing a significant decrease (*p =* 0.02) on *T6* using the SSM in JAS239-treated as compared to control animals. This indicates a reduction of permeability and perfusion of the tumour with JAS239 treatment.

No significant change or trend was noticed for the GL261 mice treated with JAS239 (Figure 5 and Appendix A).


K_ep_


No significant correlation between tumour volume and clustered K_ep_ pixels were found using any of the models (Table 1 and Table 2). In terms of mean value, no significant correlations were found between K_ep_ and tumour volume using any model, as shown in Appendix A.

For F98 rat GBMs, no significant change in $N_{\mathrm{Kep},y}$ (where y = low, med and high) was observed during treatment, as shown in Figure 6. Appendix A shows the boxplots of percentage change in the mean K_ep_ using the SSM and 2CXM models after treatment with JAS239 and saline. A significant decrease in mean K_ep_ (*p =* 0.045 and 0.02) was observed on *T3* and *T6* in JAS239 compared to control animals using SSM. A significant reduction in mean K_ep_ (*p =* 0.043 and 0.02) was observed on *T3*, and *T6* in JAS239-treated animals using the 2CXM.

No significant change or trend in any of the parameters was observed for the GL261 mice treated with JAS239.
$$v_{e}$$

A significant negative correlation between $N_{v_{e}\;,low}$ and tumour volume was found using TM (−0.43, *p*
*<*  0.01) and ETM (−0.42, *p* < 0.01), as shown in Table 1C. A significant negative correlation between $N_{v_{e}\;,med}$ and tumour volume was found using TM (−0.43, *p* < 0.01) and ETM (−0.43, *p* < 0.01) and the 2CXM (−0.52, *p* < 0.01) as shown in Table 2A. A significant positive correlation (*r* > 0.5, *p* < 0.001) between $N_{v_{e}\;,high}$ and tumour volume was found using all models as shown in Table 2B. In terms of mean value, no significant correlations were found between $v_{e}$ and tumour volume using any of the models as shown in Appendix A suggesting the utility of the cluster analysis for assessing intra-tumoural heterogeneity in extracellular-extravascular volume.

For F98 rat GBMs, a significant increase in $N_{v_{e}\;,low}$ was observed in JAS239-treated animals (*p =* 0.04) between *T3* and *T8* using 2CXM, as shown in Figure 7. A significant increase in $N_{v_{e},high}$ (*p =* 0.04) was observed on *T6* in JAS239-treated compared to control animals using 2CXM. A significant increase in mean $v_{e}$ (*p =* 0.043) was observed on *T6* in JAS239-treated as compared to control animals and a significant decrease (*p =* 0.045) between *T3* and *T8* of JAS239 treatment was found using the SSM. A significant increase in mean $v_{e}$ (*p =* 0.02 and 0.02) was observed on *T3* and *T6* in JAS239-treated as compared to control group and a significant decrease (*p =* 0.03) between *T3* and *T8* of JAS239 treatment was found using the 2CXM, as shown in Appendix A. This indicates reduced cell density with JAS239 treatment.

No significant change or trend was noticed for the GL261 mice treated with JAS239 (Figure 5 and Appendix A).
$$v_{p}$$

No significant correlation between tumour volume and clustered $v_{p}$ pixels were found using any model (Table 1 and Table 2). In terms of mean value, no significant correlations were found between $v_{p}$ and tumour volume using any of the models, as shown in Appendix A.

For F98 rat GBMs, no significant change in $N_{v_{p}\;,y}$ (where y = low, med and high) was observed during treatment, as shown in Figure 8. No significant change in mean $v_{p}$ was observed between JAS239-treated and control animals, as shown in Appendix A.

No significant change or trend was observed for the GL261 mice treated with JAS239.
$$F_{p}$$

No significant correlation between tumour volume and clustered $F_{p}$ pixels were found using any model (Table 1 and Table 2). In terms of mean value, no significant correlations were found between $F_{p}$ and increasing tumour volume using any of the models, as shown in Appendix A.

For F98 rat GBMs, a significant increase in $N_{F_{p},low}$ (*p =* 0.02) was observed on *T3* in JAS239-treated as compared to control animals using 2CXM and a statistically significant decrease in $N_{F_{p},med}$ (*p =* 0.02) and $N_{F_{p},high}$ (*p =* 0.02) was observed on *T3* in JAS239-treated compared to control animals using 2CXM, as shown in Figure 8. Mean $F_{p}$ decreased significantly (*p =* 0.028) on *T3* in JAS239-treated as compared to control animals, as shown in Appendix A. This indicates a reduction in perfusion with JAS239 treatment.

No significant change or trend was noticed for the GL261 mice treated with JAS239 (Figure 5 and Appendix A).
$$\tau_{i}$$

As shown in Table 1 and Table 2, no significant correlation between the tumour volume and clustered $\tau_{i}$ pixels were found using any of the models. In terms of mean value, no significant correlations were found between $\tau_{i}$ and tumour volume using any of the models as shown in Appendix A.

For F98 rat GBMs, no significant change in $N_{\tau_{i}\;,y}$ (where y = low, med and high) was observed during treatment, as shown in Figure 8. No significant change in mean $\tau_{i}$ was observed between JAS239-treated and control animals, as shown in Appendix A.

No significant change or trend was noticed for the GL261 mice treated with JAS239.

### 3.2. Tumour Volume

At the end of treament day (*T6*), the percent change with respect to baseline in tumour volumes in JAS239-treated animals was compared to control animals. For F98 rat GBMs, the control saline treated animals demonstrated a statistically significant increase (*p =* 0.033) in tumour volume as compared to JAS239-treated animals (Appendix A). For GL261 mice GBMs, the control saline treated mice also showed a larger increase in tumour volume as compared to JAS239-treated animals (Appendix A). However, the increase was not statistically significant.

## 4. Discussion

Using different pharmacokinetic models, our study demonstrated the utility of the clustering approach in evaluating tumour haemodynamic heterogeneity in the F98 and GL261 GBMs. A correlation between the tumour volume and clustered pharmacokinetic markers clearly demonstrated hemodynamic variations within the tumour. We further showed that the clustering algorithm could assess treatment induced changes in the F98 and GL261 GBMs.

GBMs are highly heterogeneous and hypoxic compared with other brain tumours, exhibiting considerable variation in the microvascular structure [45,47]. The spatial variations may change during tumour growth [45] or with treatment. Imaging-based biomarkers have the potential to evaluate intra-tumoural heterogeneity and its relationship to tumour growth and response to therapy [26,48]. MRI is a useful modality for evaluating spatial and temporal variations in alterations in the biologic characteristics of tumours that may include changes in apoptosis, cellular proliferation, cellular invasion, and angiogenesis [49]. If MRI features of the tumour correlate with genetic characteristics, it may be possible to noninvasively identify tumour genetic features [49]. Contrast enhancement on DCE-MRI results from the breakdown of the blood–brain barrier and can be used to identify areas of necrosis. In addition, physiologic characteristics such as apparent diffusion coefficient and perfusion have been found to correlate to tumoural cellularity and angiogenesis [49]. The spatial distribution of DCE-MRI-based perfusion parameters within the tumour area is a very important tool allowing for the assessment of the angiogenic compositions of the tumour [22,23]. The profound intra-tumour vascular heterogeneity in GBMs is due to aberrant microvasculature and inefficient nutrient delivery [50]. In this study, we demonstrated segregation of the heterogeneous regions of the GBM, including the existence of necrotic regions with the help of DCE-MRI-based pharmacokinetic parameters. As the tumour volume increased, a significant increase in the number of necrotic pixels was observed, indicating a decrease in vascular density and an increase in *de novo* cell death. This is because the tumour parenchyma outgrows the vascular network [51]. Few studies have also reported that DCE-MRI can be used to assess intra-tumour heterogeneity [52,53] and that a variation in $K^{trans}$ between and within tumours is not related to tumour size [52]. When the mean value of the parameters $K^{trans}$, $v_{e}$, K_ep,_ $v_{p}$_,_ $\tau_{i}$_,_
$F_{p}$ were used, we also did not observe any significant correlation with tumour volume. This suggests that the mean values fail to reflect tumour heterogeneity, which is an essential feature for assessing treatment since it can identify the sub population of cells that are underperfused and likely to be treatment resistant.

Automatic segmentation into informative subregions (“habitats”) within tumours can be linked to underlying tumour pathophysiology [26]. Deep learning methods do not require accurate segmentation; it creates their features through multiple layers of learning [54]. For medical images, convolutional neural networks (CNNs) and unsupervised methods are commonly used for dividing data into groups, or clusters, with similar properties. Clustering of the image voxels has been done before based on the pharmacokinetic parameters [55]. We used a similar hierarchical clustering to assess tumour haemodynamic heterogeneity without introducing a bias typically introduced while using an ROI-based approach.

Correlation analysis was performed to assess the relationship between the number of pixels representing a cluster in the DCE parametric maps with tumour growth. Although we did not find any significant correlation between tumour growth and $N_{K^{trans},y}$ using any of the models, we did observe a significant correlation between $N_{v_{e},y}$ and tumour growth. A significant increase in the number of necrotic pixels, a significant negative correlation between $N_{v_{e}\;,low}$ and a significant positive correlation between $N_{v_{e}\;,high}$ with increasing tumour volume indicates decreased vascular density with tumour growth. Based on Pearson’s correlation coefficient and R^2^, the SSM and the 2CXM models were selected to detect changes in $K^{trans}$, $v_{e}$, K_ep,_ $v_{p}$_,_ $\tau_{i}$_,_
$F_{p}$ in the tumours with JAS239 treatment.

The relationship between DCE-MRI parameters (essentially $K^{trans}$, which reflects the effectiveness of the delivery of oxygen and therapeutic agents between the plasma and interstitial space of the tumour) and clinical outcome following treatment has been evaluated before [56]. Studies on hepatocellular and renal cancer suggest that both a higher $K^{trans}$ at baseline and an early reduction of $K^{trans}$ are significantly associated with improved outcomes following anti-angiogenic treatment [57,58,59,60,61]. Decrease in permeability has been found to be predominant response of tumour vasculature to bevacizumab therapy in GBMs [62]. GBM patients who underwent DCE-MRI prior and up to 96 h after initialisation of bevacizumab (BEV) treatment showed early reduction in $K^{trans}$ (measured 96 h after treatment initialisation) but did not correlate with overall survival. The extent of early reduction in $K^{trans}$ following treatment initialisation with BEV and dose-intense temozolomide did not have an impact on disease outcome in recurrent GBM [63]. However, another study in patients with recurrent GBM showed that a greater reduction in $K^{trans}$ was associated with a significantly longer overall survival [64]. Human renal cell carcinoma xenograft models showed temporal changes following treatment with sorafenib. $K^{trans}$ was significantly decreased compared with baseline values as early as three days after the start of sorafenib [65]. $v_{e}$ has been proven clinically important in assessing tumour response to treatment [66]. Increase in $v_{e}$ at an early stage of chemoradiotherapy in cervical cancer patients has also been reported [66].

In this study, we also observed early changes in pharmacokinetic parameters in response to treatment with JAS239 in F98 rat GBMs. A significant decrease in$\;N_{K^{trans},high}$ (using SSM) and $N_{F_{p},high}$ (using 2CXM) along with a significant increase in $N_{K^{trans},low}$ (using SSM) and $N_{F_{p},low}$ (using 2CXM) during treatment (*T3*) suggests therapy induced acute changes in tumour permeability and perfusion. The change in $F_{p}$ in the present study suggests that tumour perfusion, apart from endothelial permeability, is critical to delivering therapeutic agents. A significant decrease in$\;N_{K^{trans},high}$ using the 2CXM model and mean $K^{trans}$ using SSM on *T6* with JAS239 suggests a sustained decrease in permeability and perfusion until the end of treatment. A significant increase in $N_{v_{e},high}$ using 2CXM on *T6* suggests reduced cell density with JAS239 treatment and could be theoretically interpreted as increase in extracellular fluid levels due to the disintegration of tumour cell membranes. K_ep_ calculated as a ratio between $K^{trans}$ and $v_{e}$ did not provide any unique information, but a decrease in mean K_ep_ was observed with JAS239 treatment. No significant change in $v_{p}$ was observed which is somewhat surprising because new blood vessel proliferation in GBM results in increased vascular density. This will require further investigation with larger sample size and histological validation.

Although $\tau_{i}$ has been suggested to be a reliable marker of metabolic activity [16], we did not observe any significant changes in this parameter with ChoK inhibition. Although we observed treatment response in terms of early changes in reduction of permeability and perfusion along with reduced cell density with treatment in the F98 cases, no significant differences in these parameters was found for the GL261 tumour, indicating that the GL261 model may be highly resistant to treatment with regards to changes in vascular parameters. These findings agree with a previous study [31], where an anti-vascular agent (BEV) was used for treatment in this model. Response to treatment is associated with change in tumour volume [67,68,69,70]. Both the tumour models demonstrated a trend in reduction of tumour volumes with treatment with a significant reduction in the F98 cases.

Our results also demonstrated that the overall trend in percentage change of the pharmacokinetic parameters followed the same pattern using SSM and 2CXM, which were chosen for monitoring treatment response based on high mean R^2^ in the tumour ROI and a significant increase in the number of necrotic pixels with an increase in tumour volume along with a decrease in high $K^{trans}$ pixels with increase in tumour volume, suggesting that these models provide similar assessment of the haemoynamic parameters.

### Limitations

The significance levels in various parameters were different, which may have been due to the lower number of animals for the treatment studies. Some technical limitations of this work include analysing data with a fixed T1 value for which the information was obtained from a small subset (due to scan time constraints) out of the total number of F98 rat GBMs used. Moreover, due to scan time constraints, *B_1_* mapping for bias correction in estimation of the variable flip angle based T1 maps was not performed in this study (for GL261 mouse GBMs). In addition, the use of population AIF does not consider the individual variations and might lead to biased estimation of the parameters. Another limitation of the study is the low number of animals used for monitoring the treatment response and the absence of histopathological results to confirm the imaging findings. Correlating the findings from MRI with histopathological data can provide better confirmation of the findings. Future studies will focus on deteremining an optimal combination of the pharmacokinetic models to fit all the tumour pixels. In addition, we plan to perform the clustering approach based on the optimal combination of these models for better accuracy in identifying tumour haemodynamic heterogeneity.

## 5. Conclusions

In conclusion, this study demonstrates that the clustered DCE-MRI-based pharmacokinetic parameters generated using different models correlate with tumour volume and may be used for detecting and assessing early therapeutic response. The 2CXM and SSM models were found to be best to monitor treatment response.

## Figures and Tables

**Figure 1 cancers-14-01223-f001:**
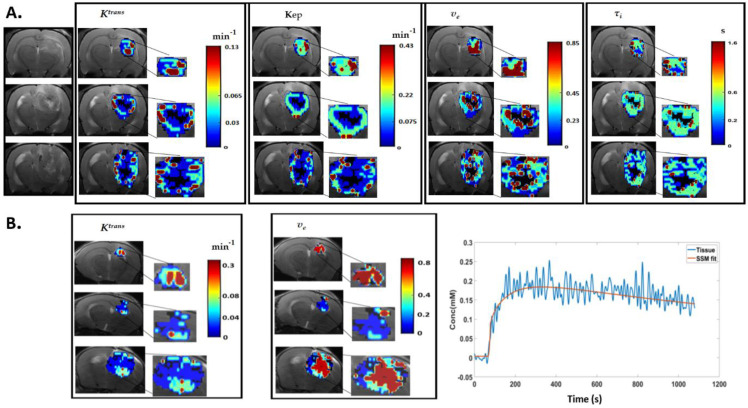
(**A**) Assessing tumour haemodynamic heterogeneity with DCE-MRI using the SSM model. T2 weighted images from a control animal bearing F98 tumour at *T0* (top row), *T6* (middle row), and *T8* (bottom row). The colour coded parametric maps of $K^{trans}$, K_ep,_ $v_{e}$_,_ $\tau_{i}$ and segmented into clusters using hierarchical clustering algorithm are displayed as overlays on the T2 weighted images in the respective columns, and the zoomed area of the tumour are shown. The poor fit necrotic clusters (black), low (blue), medium (green), and high value clusters (red) are shown, and the mean values of the parameters in the respective clusters are explained in the colour bars. (**B**) Clustered $K^{trans}$ and $v_{e}$ map using SSM overlaid on T2 weighted images from time points *T0*, *T6*, *and T10* from a GL261 mouse GBM and an example of a GL261 tumour tissue Gd-DTPA kinetics and corresponding model fitting using SSM.

**Figure 2 cancers-14-01223-f002:**
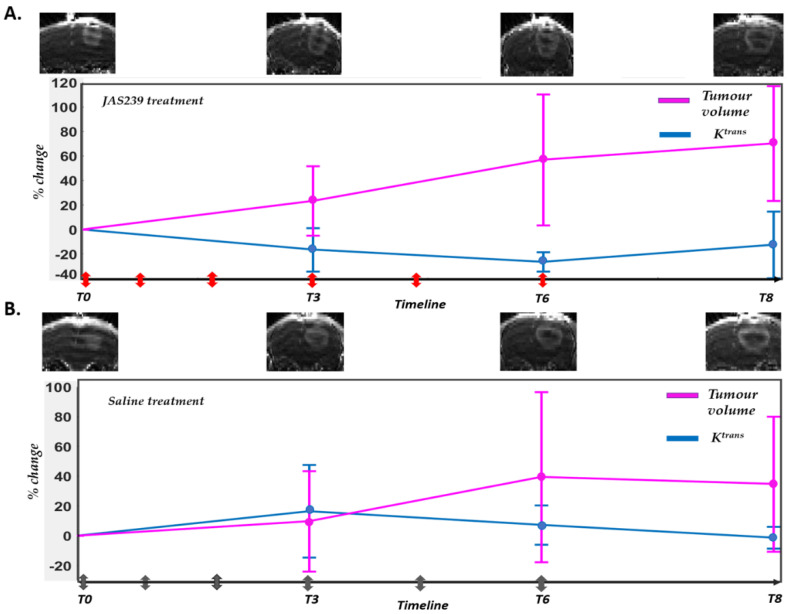
(**A**) T2 weighted images from a JAS239-treated rat acquired on *T0*, *T3*, *T6*, and *T8*. Percentage change (with respect to baseline) in volume and mean $K^{trans}$ (obtained using SSM) at respective time points along with points of injections marked with arrows (red) (**B**) T2 weighted images from a saline treated (control) rat acquired on *T0*, *T3*, *T6*, and *T8*. Percentage change in volume and mean $K^{trans}$ (obtained using SSM) at respective time points along with points of injections marked with arrows (grey).

**Figure 3 cancers-14-01223-f003:**
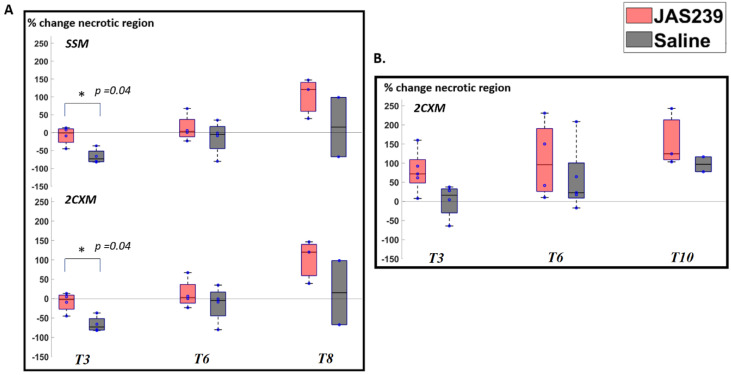
(**A**) Box plot for percentage change (with respect to baseline) in necrotic pixels in F98 rat GBM with JAS239 (red) and saline (grey) treatment at different time points using the SSM and 2CXM models, respectively. (**B**) Box plots showing percentage change (with respect to baseline) in GL261 mice GBM with regards to the number of pixels representing the necrotic pixels. The asterisk indicates significant difference.

**Figure 4 cancers-14-01223-f004:**
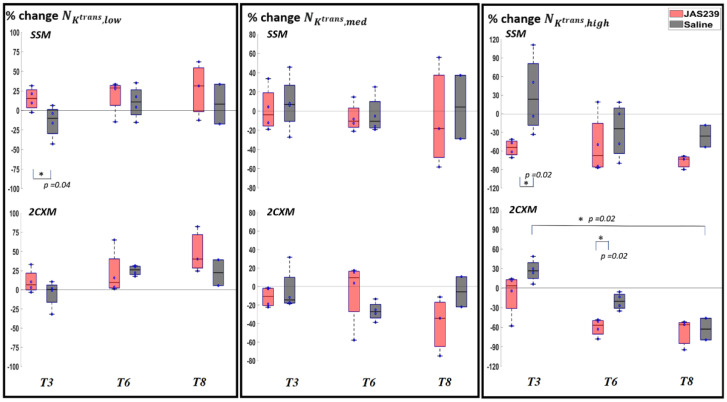
Box plot for percentage change (with respect to baseline) in $N_{K^{trans},y}$ (where y = low, med and high) in F98 rat GBM with JAS239 (red) and saline (grey) treatment at different time points using the SSM (top) and 2CXM (bottom) models, respectively. The asterisk indicates significant difference.

**Figure 5 cancers-14-01223-f005:**
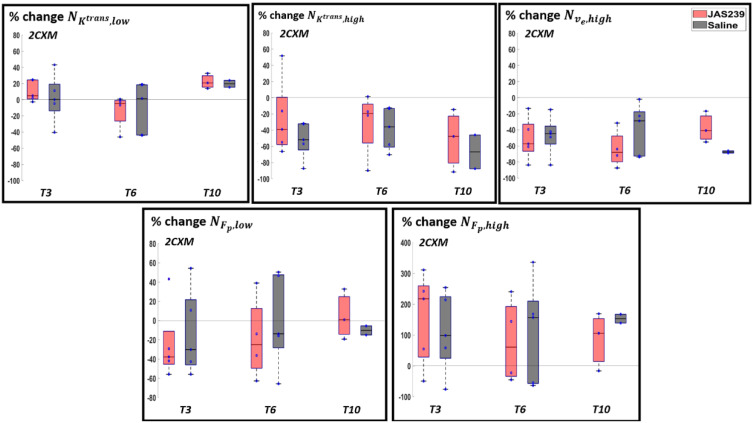
Box plots showing percentage change (with respect to baseline) in GL261 mice GBM with regards to the number of pixels representing the clustered parameters $K^{trans}$ and $v_{e}$ on the top and $F_{p}\;$on the bottom (only the parameters that changed significantly from baseline values are shown).

**Figure 6 cancers-14-01223-f006:**
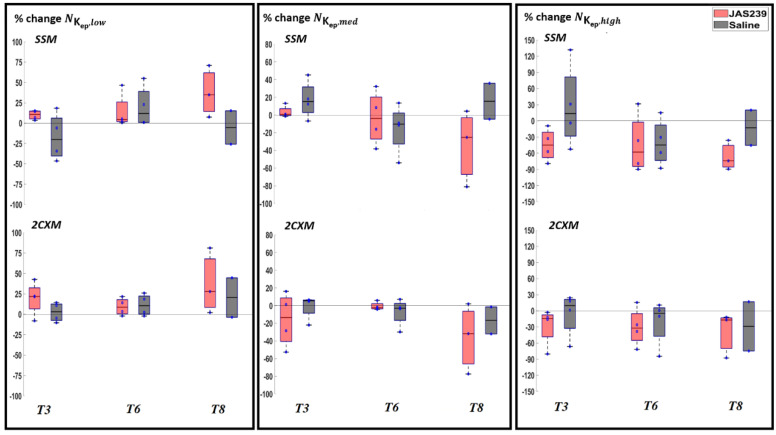
Box plot for percentage change (with respect to baseline) in $N_{\mathrm{Kep},y}$ (where y = low, med and high) in F98 rat GBM with JAS239 (red) and saline (grey) treatment at different time points using the SSM (top) and 2CXM (bottom) models, respectively.

**Figure 7 cancers-14-01223-f007:**
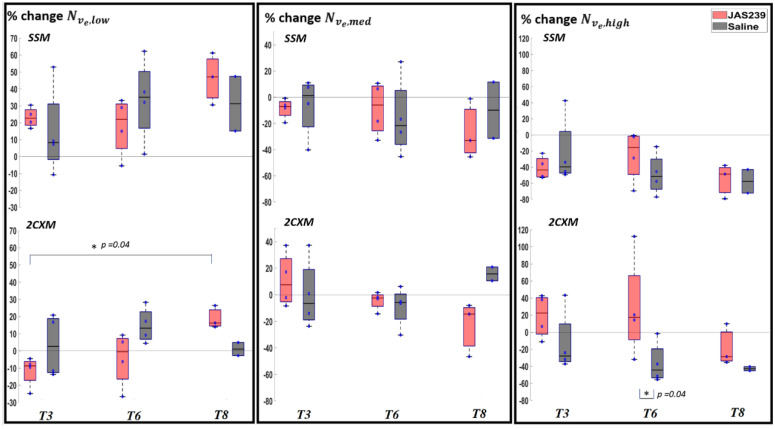
Box plot for percentage change (with respect to baseline) in $N_{v_{e}\;,y}$ (where y = low, med and high) in F98 rat GBM with JAS239 (red) and saline (grey) treatment at different time points using the SSM (top) and 2CXM (bottom) models, respectively. The asterisk indicates significant difference.

**Figure 8 cancers-14-01223-f008:**
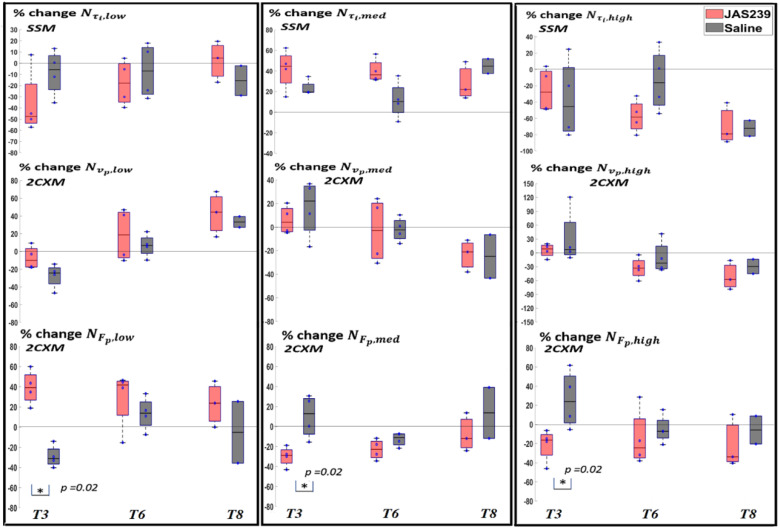
Box plot for percentage change (with respect to baseline) in $N_{\tau_{i}\;,y}$ (where y = low, med and high) (**top**) in F98 rat GBM with JAS239 (red) and saline (grey) treatment at different time points using the SSM model. The differences in the $N_{v_{p}\;,y}$ using 2CXM model are shown in the middle and differences in the $N_{F_{p}\;,y}$ using 2CXM model are shown in the bottom. The asterisk indicates significant difference.

**Table 1 cancers-14-01223-t001:** Pearson’s correlation (*r*) for tumour volume vs. necrotic pixels (A), $R^{2}\;$for fitted pixels across all the datasets using all models (B) and Pearson’s correlation (*r*) for tumour volume vs. $N_{x,low}$ (C).

	A: Pearson’s Correlation (r) for Tumour Volume vs. % of Necrotic Pixels				
	TM	ETM	SSM	2CXM	SSM2
	−0.23	−0.23	**0.53 (*p* < 0.001)**	**0.54 (*p* < 0.001)**	0.24
	B: Mean ± S.D $R^{2}\;$for fitted pixels across all the datasets using all models				
	TM	ETM	SSM	2CXM	SSM2
$R^{2}\;$	0.95 ± 0.01	0.956 ± 0.01	0.962 ± 0.007	0.968 ± 0.007	0.96 ± 0.008
	C: Pearson’s correlation (r) for tumour volume vs. $N_{x,low}$				
	TM	ETM	SSM	2CXM	SSM2
$N_{K^{trans},low}$	0.008	−0.006	0.26	0.25	0.18
$N_{\mathrm{Kep},low}$	−0.07	0.11	0.26	0.16	0.17
$N_{v_{e},low}$	**−0.43 (*p* < 0.01)**	**−0.42 (*p* < 0.01)**	−0.08	−0.18	0.09
$N_{v_{p},low}$	N.A	0.3	N.A	0.3	0.28
$N_{\tau_{i},low}$	N.A	N.A	0.20	N.A	0.1
$N_{F_{p},low}$	N.A	N.A	N.A	0.003	N.A

Note. TM, Tofts model; ETM, extended Tofts model; SSM, Shutter speed model; 2CXM, two-compartment exchange model; SSM2, second generation Shutter speed model; N.A, not applicable. Bold in the text indicates significant result.

**Table 2 cancers-14-01223-t002:** Pearson’s correlation (*r*) for tumour volume vs. $N_{x,med}$ (A), and Pearson’s correlation (*r*) for tumour volume vs. $N_{x,high}$ (B).

	A: Pearson’s Correlation (r) for Tumour Volume vs. % of $N_{x,med}$				
	TM	ETM	SSM	2CXM	SSM2
$N_{K^{trans},med}$	−0.01	−0.03	−0.12	−0.27	−0.06
$N_{\mathrm{Kep},med}$	−0.03	−0.01	−0.24	−0.21	−0.20
$N_{v_{e},med}$	**−0.43 (*p* < 0.01)**	**−0.43 (*p* < 0.01)**	−0.27	**−0.52 (*p* < 0.01)**	−0.30
$N_{v_{p},med}$	N.A	−0.3	N.A	−0.15	−0.4
$N_{\tau_{i},med}$	N.A	N.A	−0.1	N.A	0.05
$N_{F_{p},med}$	N.A	N.A	N.A	−0.31	N.A
	B: Pearson’s correlation (r) for tumour volume vs. % of $N_{x,high}$				
	TM	ETM	SSM	2CXM	SSM2
$N_{K^{trans},high}$	−0.02	−0.05	−0.25	−0.22	−0.11
$N_{\mathrm{Kep},high}$	0.01	−0.14	−0.21	−0.07	−0.03
$N_{v_{e},high}$	**0.65 (*p* < 0.001)**	**0.65 (*p* < 0.001)**	**0.7 (*p* < 0.001)**	**0.75 (*p* < 0.001)**	**0.6 (*p* < 0.001)**
$N_{v_{p},high}$	N.A	−0.25	N.A	−0.3	−0.1
$N_{\tau_{i},high}$	N.A	N.A	−0.2	N.A	−0.2
$N_{N_{F_{p},high}}$	N.A	N.A	N.A	0.24	N.A

Note. TM, Tofts model; ETM, extended Tofts model; SSM, Shutter speed model; 2CXM, two-compartment exchange model; SSM2, second generation Shutter speed model; N.A, not applicable. Bold in the text indicates significant result.

## Data Availability

The data presented in this study are available on request from the corresponding author.

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
