# Peer review of "Assessing Tumour Haemodynamic Heterogeneity and Response to Choline Kinase Inhibition Using Clustered Dynamic Contrast Enhanced MRI Parameters in Rodent Models of Glioblastoma"

_cancers, 2022, doi:10.3390/cancers14051223_

Round 1

Reviewer 1 Report

The authors present a very complex study aimed at demonstrating the existence of tumor hemodynamic heterogeneity in rodent models of glioblastoma using the DCE MRI technique.

They also speculate a different response to choline kinase inhibitor between the heterogenic and non-heterogenic groups.

The study is potentially interesting, but some major revisions are needed as follows:

  • The introduction should be shortened. It should focus uniquely on the advantages coming from the DCE-MRI technique; 
  • The level of complexity of the DCE data analysis reported is incompatible with a clinical journal. The various equation should be eventually reported as supplement material; Stay focused on the clinical relevance of the technique; 
  • A large part of the discussion stresses the key role of the reported study in characterizing the tumor’s aggressiveness based on the heterogeneity of the vascular supply. This makes no sense for GBMs. While admitting the existence of a hemodynamic heterogeneity within the tumor volume, the pieces of evidence about a better response of the less heterogenic tumors to therapy are still lacking. Accordingly, some sentences cannot be accepted. Please revise the discussion. 
  • A further point regards the choice of the therapeutic agent and the rationale of the study. Why, in the light of a detected hemodynamic heterogeneity of the tumor, did the authors decide to use JAS239 treatment rather than an antiangiogenic drug as bevacizumab? It would be made more sense for the clinical practice.

Author Response

Response to the editor and reviewers

We would like to thank the editor and reviewer for their effort and time in reviewing our manuscript. They have provided us with helpful suggestions and remarks, which we believe have helped us improve the quality of this manuscript. All the comments and suggestions have been addressed and incorporated in the revised version (using track changes), as indicated in the point-by-point response below .

Reviewer 1:

Comments:

  1. The introduction should be shortened. It should focus uniquely on the advantages coming from the DCE-MRI technique; 

REPLY: We have shortened the introduction as suggested. We have removed the sections about DSC imaging and have now focused on the advantages of DCE imaging as a perfusion weighted imaging modality as well as derived pharmacokinetic parameters.

  1. The level of complexity of the DCE data analysis reported is incompatible with a clinical journal. The various equation should be eventually reported as supplement material; Stay focused on the clinical relevance of the technique; 

REPLY: Thanks for the comment. We have now moved section 2.2. DCE data analysis as supplement material section A1.

  1. A large part of the discussion stresses the key role of the reported study in characterizing the tumor’s aggressiveness based on the heterogeneity of the vascular supply. This makes no sense for GBMs. While admitting the existence of a hemodynamic heterogeneity within the tumor volume, the pieces of evidence about a better response of the less heterogenic tumors to therapy are still lacking. Accordingly, some sentences cannot be accepted. Please revise the discussion.

REPLY: Whilst we agree that GBM are highly aggressive and heterogenous to begin with, we strongly believe that assessing tumour hemodynamic heterogeneity, as presented in this study, does not only include the vascular features of perfusion and permeability, but also extracellular volume, half-life of the intracellular water ( ) and other parameters. These physiological parameters play a very important role in the growth of the tumour and may also be responsible in its response/resistance to therapies. We would like to emphasize that we are not trying to quantify the aggressiveness of the tumour based on heterogeneity. Rather, the purpose of this work was to assess the tumour heterogeneity in terms of variations in the DCE-MRI derived pharmocokinetic parameters in the tumour region and clustering them into pixels containing high, medium, low values as well as necrosis. We observed that with change in tumour volume, the distribution of some of these clustered parameters in the tumour region varied. Also, with treatment, the distribution of some of these clustered parameters followed a certain trend of increment or reduction. Please note that the study is not so much  about assessment of aggressiveness of the tumours, since all these tumours were highly aggressive GBM, but assessing heterogeneity with different sizes depending on the day when it was scanned. Also, we cannot quantify heterogeneity as we do not have a proper reference to measure this against. Considering that all tumours are heterogenous, we simply measured the change in distribution of the high, med, low valued DCE-MRI parameters and also necrotic areas with change in volume and with treatment. This can help to check the response to treatment in terms of change in these parameters and also with change in tumour growth.

We admit that some of our text may have been misleading. As such, we have modified the discussion by eliminating the sentence “  and  have been shown to be sensitive parameters in distinguishing between tumour aggressiveness [46-48]” in the revised version of this manuscript. The revised discussion section mostly focuses on the correlation between the number of pixels representing a cluster in the DCE parametric maps and tumour growth. It also discusses the change in DCE-MRI parameters following treatment. We also added a part discussing change in tumour volume following treatment in lines 463-465.

  1. A further point regards the choice of the therapeutic agent and the rationale of the study. Why, in the light of a detected hemodynamic heterogeneity of the tumor, did the authors decide to use JAS239 treatment rather than an antiangiogenic drug as bevacizumab? It would be made more sense for the clinical practice.

REPLY: The rationale for studying the effects of JAS239 was stated in the introduction and we apologize if that was not clear in itself. What we are trying to do in this study is to evaluate the role of different DCE-MRI derived parameters in understanding the tumour heterogeneity and how these parameters can be used to assess treatment response. This therapeutic could be any and does not have to be specific to alteration of tumour vasculature, something like bevacizumab does as the reviewer suggests. The reason for us to choose JAS239 as a potential novel drug in the treatment of GBM was that it and its non-fluorescent analogue MN58b have recently been used to assess therapeutic response in preclinical models of breast and brain tumours. However, those studies used MR spectroscopy, which is highly specific but less sensitive. The use of MRI measures on the other hand provides additional information at a much higher resolution, which is what we have demonstrated. JAS239 inhibits ChoK intracellularly, preventing choline phosphorylation, and has been shown to induce tumour growth arrest and cell death in a breast cancer model, as reported in references [24, 25]. The inhibition of ChoK may also sensitise GBMs to chemotherapy and radiotherapy [23].  GBMs are more hypoxic compared with other brain tumours exhibiting considerable variation in the microvascular structure. Although inhibition of ChoK has been shown to have therapeutic effects, the relationship of perfusion to hypoxia is not well understood. Antiangiogenic drug as bevacizumab has already been already tested in GBM as mentioned in references [56, 57] and an early reduction of  with treatment was reported in these studies. Also, no significant differences in  was found for the GL261 tumour according to the study [27], where bevacizumab was used for treatment.  

Reviewer 2 Report

Comments and Suggestions for Authors

The Manuscript is interesting for readers of Cancers in terms of preclinical GBM therapy monitoring.

The aim of the study was to assess hemodynamic heterogeneity in GBM rodent models. The authors did this via assessment of different perfusion parameters (ktrans, ??, ??, ?? , Kep, ??) derived from DCE MRI, using different fitting models.

Their second aim was to monitor changes in tumour haemodynamic heterogeneity of these markers in response to treatment with the ChoK Inhibitor JAS239.

Perfusion Imaging is of high interest in glioma MR imaging in clinical routine, and the preclinical study therefore of interest.

The paper is well written and logically constructed.

However, the authors need to clarify the following questions before the acceptance of manuscript for publication.

Introduction:

Please mention that in clinical neuro-mr-imaging a leakage correction can be performed (line 59, after Ref 6).

Please search a Ref where DCE MRI has been used for clinical brain tumor imaging in patients, at least where k trans derived from DSC MRI has been used (if no other ref available please check: NeuroImage: Clinical 20 (2018) 51–60, https://doi.org/10.1016/j.nicl.2018.07.001)

Please add a Ref that the degree of heterogeneity in vascular permeability and nutrient consumption in tumor microenvironment correlates with aggressiveness of the tumor.

“Changes in tumor microenvironment leads..” Please check spelling. Please check word order / wording: …”other models including the shutter speed (SSM) and two-compartment exchange model (2CXM) models, can provide additional physiologically sensitive parameters like ? and ? .”

Has JAS239 also been used in gliomas? Please add a Ref.

Materials and Methods:

… “from all GL261 mouse GBMs to obtain pre-contrast T1 maps”: Do you mean :…from all GL261 GBM bearing mouse brains?

Check wording: …”20x20x4.8 mm3 for (for mice),” …

Add the protocol of the higher resolution T2 weighted sequence.

The tumour  haemodynamic heterogeneity was evaluated depending on tumour volumes… What do you mean? Do you mean on basis of tumour volumes? In the area of T2 visible signal alterations?

ROI for AIF in superior sagittal sinus: I suppose this is because perfusion is very fast in rodents?

The assessment of Tumour volumes is not clearly described.

Please add a Ref where it is described that response to treatment is associated with increase in tumour volume.

Please describe if you also saw shrinkage of tumour volume in any of the cases.

Please include a definition of R2 and the reason for using 0.7 as a cut-off.

Results:

Fig 2 A and B: please change colour of arrows: red in A for therapy, grey in B for saline.

Accordingly colours in Fig 3. and in the rest of the Figures.

It would be easier to read if Fig 5 f was put next to or under Fig 3.

Fig 5: For coherence please mention model used in Fig 5.

It would be more coherent to put Fig 5a (??) as 5c after k trans high.

Please check Fig S3 and Fig 1A for consistency regarding ??.

Kep: Please add cross reference to Fig S2.

Figure 8: Please check legend.

?? : Please check spelling in the following paragraph.

Discussion:                

Please try to find an additional more recent ref on early reduction of ktrans after therapy onset.

What do you think is the reason for non-significant reduction in permeability in GL261 mice GBM trated with JAS239? Did the lower sample size play a role?

Please add a heading for Limitations.

Check wording of first sentence of Limitations: “The significance levels in various parameters were different, which may have been due to 500 the lower number of animals for the treatment studies.”

Conclusion:

Please move the last 2 sentences to the limitation section as an outsight on further studies that should be done in the future for even more exact results.

Refs:

Ref 22: Please check format and add year of publication.

Additional:

Please provide the IRB number and ClinicalTrial.gov registered code (NCT number).

Round 2

Reviewer 1 Report

Points 3 and 4 have not been addressed adequately by the authors.

The entire study lies on the assumption that what the authors indicate as “heterogeneity” of the tumor should be in whatever manner correlated with the biological behavior of the GBM. While agreeing on a certain role on the tumor vasculature on the growth and recurrence, the details of this relationship are still to date unclear. The literature reports conflicting data about whether we should do consider “more malignant” a necrotic glioblastoma or a densely vascularized one. With any probability, the authors espouse the former position. But this point ought to be been clearly stated since the Introduction and stressed in the Discussion. The authors added the references n. 18 and 19 which are not pertinent to glioblastoma. Please note that neuroepithelial malignant tumors, and especially glioblastoma, hold known biological pathways of immune escape completely different from those of other non-CNS neoplasms. By the reason of the lacking of definitive data delineating these relationships, the simple term “heterogeneity” cannot be accepted as a concept on which to base the assessment of response to any therapy, including JAS239. Unfortunately, these aspects are still unclear in this manuscript.

About point 4, what was suggested was to simply specify the reasons for the choice of JAS239 in comparison with other drugs. While embracing with enthusiasm the novelty of testing it for GBM, no reference has been listed in the reference list about the use of JAS239 in GBM. This is the main reason why the no obvious choice of JAS239 in the present study should have been justified.

Author Response

Response to the editor and reviewers (Round 2)

We would like to thank the editor and reviewers for their effort and time spent on reviewing our manuscript. They have provided us helpful suggestions and remarks, which we believe improved the quality of this manuscript. All of the comments and suggestions have been addressed and incorporated in the revised version (using track changes), as indicated in this detailed question-answer (Q-A) rebuttal.

Reviewer 1:

Comments:

  1. The Points 3 and 4 have not been addressed adequately by the authors.

The entire study lies on the assumption that what the authors indicate as “heterogeneity” of the tumor should be in whatever manner correlated with the biological behavior of the GBM. While agreeing on a certain role on the tumor vasculature on the growth and recurrence, the details of this relationship are still to date unclear. The literature reports conflicting data about whether we should do consider “more malignant” a necrotic glioblastoma or a densely vascularized one. With any probability, the authors espouse the former position. But this point ought to be been clearly stated since the Introduction and stressed in the Discussion. The authors added the references n. 18 and 19 which are not pertinent to glioblastoma. Please note that neuroepithelial malignant tumors, and especially glioblastoma, hold known biological pathways of immune escape completely different from those of other non-CNS neoplasms. By the reason of the lacking of definitive data delineating these relationships, the simple term “heterogeneity” cannot be accepted as a concept on which to base the assessment of response to any therapy, including JAS239. Unfortunately, these aspects are still unclear in this manuscript.

REPLY: We regret to note that the reviewer thinks we have not addressed points # 3 and 4 in the revised manuscript. We agree with the reviewer that heterogeneity of the tumor is not always correlated with the biological behavior of the GBM and that the specific molecular events and pathways leading to GBM remain unclear. However, we would like to emphasize that in this study, we have taken a commonly known and well accepted feature of GBMs being highly heterogenous with areas of  necrosis and the fact that within gliomas, necrosis and heterogeneity is associated with the tumor malignancy,  prognosis as well as response/resistance to therapies.

In order to further enhance the clarity of the introduction and provide a clearer rationale, we have removed references n. 18 and 19 and added the following information :

Introduction  (lines 76 – 82)

GBMs are formed by tumour cells which differ in their morphology, genetics and biological behaviour [18, 19]. They are typically heterogeneous both on genetic and histo-pathological levels with intratumoural spatial variation in the cellularity, angiogenesis, extravascular extracellular matrix, and areas of necrosis [20, 21]. The  heterogeneity can be with regards to cellular morphology, gene expression, metabolism, angiogenic and proliferative potential, some of which can be investigated using DCE-MRI for example, in investigating angiogenic processes [22, 23].

Discussion  (lines 394 – 408)

MRI is a useful modality for evaluating spatial and temporal variations in alterations in the biologic characteristics of tumours that may include changes in apoptosis, cellular proliferation, cellular invasion, and angiogenesis [49]. If MRI features of the tumour correlate with genetic characteristics, it may be possible to noninvasively identify tumour genetic features [49]. Contrast enhancement on DCE-MRI results from the breakdown of the blood-brain barrier and can be used to identify areas of necrosis. In addition, physiologic characteristics such as apparent diffusion coefficient and perfusion have been found to correlate to tumoural cellularity and angiogenesis [49]. The spatial distribution of DCE-MRI based perfusion parameters within the tumour area is a very important tool allowing for the assessment of the angiogenic compositions of the tumour [22, 23]. The profound intra-tumour vascular heterogeneity in GBMs is due to aberrant microvasculature and inefficient nutrient delivery [50]. In this study, we demonstrated segregation of the  heterogeneous regions of the GBM, including the existence of necrotic regions with the help of DCE-MRI based pharmacokinetic parameters.

References:

  1. Bonavia R, Inda MM, Cavenee WK, Furnari FB. Heterogeneity maintenance in glioblastoma: a social network. Cancer Res. 2011 Jun 15;71(12):4055-60. doi: 10.1158/0008-5472.CAN-11-0153. Epub 2011 May 31. PMID: 21628493; PMCID: PMC3117065.
  2. Inda MM, Bonavia R, Seoane J. Glioblastoma multiforme: a look inside its heterogeneous nature. Cancers (Basel). 2014 Jan 27;6(1):226-39. doi: 10.3390/cancers6010226. PMID: 24473088; PMCID: PMC3980595.
  3. Davnall F, Yip CS, Ljungqvist G, Selmi M, Ng F, Sanghera B, Ganeshan B, Miles KA, Cook GJ, Goh V. Assessment of tumor heterogeneity: an emerging imaging tool for clinical practice? Insights Imaging. 2012 Dec;3(6):573-89. doi: 10.1007/s13244-012-0196-6. Epub 2012 Oct 24. PMID: 23093486; PMCID: PMC3505569.
  4. Vartanian A, Singh SK, Agnihotri S, Jalali S, Burrell K, Aldape KD, Zadeh G. GBM's multifaceted landscape: highlighting regional and microenvironmental heterogeneity. Neuro Oncol. 2014 Sep;16(9):1167-75. doi: 10.1093/neuonc/nou035. Epub 2014 Mar 18. PMID: 24642524; PMCID: PMC4136895.
  5. Just N. Improving tumour heterogeneity MRI assessment with histograms. Br J Cancer. 2014 Dec 9;111(12):2205-13. doi: 10.1038/bjc.2014.512. Epub 2014 Sep 30. PMID: 25268373; PMCID: PMC4264439.
  6. Walsh JJ, Parent M, Akif A, Adam LC, Maritim S, Mishra SK, Khan MH, Coman D, Hyder F. Imaging Hallmarks of the Tumor Microenvironment in Glioblastoma Progression. Front Oncol. 2021 Aug 26;11:692650. doi: 10.3389/fonc.2021.692650. PMID: 34513675; PMCID: PMC8426346.
  7. Chow D, Chang P, Weinberg BD, Bota DA, Grinband J, Filippi CG. Imaging Genetic Heterogeneity in Glioblastoma and Other Glial Tumors: Review of Current Methods and Future Directions. AJR Am J Roentgenol. 2018 Jan;210(1):30-38. doi: 10.2214/AJR.17.18754. Epub 2017 Oct 5. PMID: 28981352.
  8. Li C, Yan JL, Torheim T, McLean MA, Boonzaier NR, Zou J, Huang Y, Yuan J, van Dijken BRJ, Matys T, Markowetz F, Price SJ. Low perfusion compartments in glioblastoma quantified by advanced magnetic resonance imaging and correlated with patient survival. Radiother Oncol. 2019 May;134:17-24. Doi: 10.1016/j.radonc.2019.01.008. Epub 2019 Jan 31. PMID: 31005212; PMCID: PMC6486398.

  1. About point 4, what was suggested was to simply specify the reasons for the choice of JAS239 in comparison with other drugs. While embracing with enthusiasm the novelty of testing it for GBM, no reference has been listed in the reference list about the use of JAS239 in GBM. This is the main reason why the no obvious choice of JAS239 in the present study should have been justified.

REPLY: We apologize if this point was not clear in our previous answer. The effectiveness of JAS239 on a GBM model is yet to be reported, which was the basis of this manuscript. Thus, we could not add a reference about the use of JAS239 in GBM. However,  other ChoK inhibitors  have been developed and tested across different cancer types, including GBM, and have shown in vitro and in vivo efficacy. Until the development of novel JAS239 ChoK inhibitor, MN58b was thought to be the most potent ChoK inhibitor that has been tested pre-clinically. Treatment efficacy has been demonstrated in the highly necrotic and aggressive F98 model with ChoK inhibitor MN58b by Kumar et al.(ref :27). Some preliminary work on effectiveness of JAS239 in GBMs has been presented by our research group in the ISMRM conference (Kelly et al, Effect of hypoxia and choline kinase inhibitor on choline metabolism of brain tumour cells using high resolution ¹H NMR. In Proceedings of the 2020 ISMRM virtual conference. Poster no: 4815 and Kelly et al, MR spectroscopy to assess decreased tumor choline as a marker of response to choline kinase inhibitors. In Proceedings of the 2019 ISMRM conference. Abstract #2392). However, according to the journal guidelines, abstract presentations at scientific meetings cannot be referenced.
